# In Vitro and Ex Vivo Kinetic Release Profile of Growth Factors and Cytokines from Leucocyte- and Platelet-Rich Fibrin (L-PRF) Preparations

**DOI:** 10.3390/cells11132089

**Published:** 2022-06-30

**Authors:** Xuzhu Wang, Melissa R. Fok, George Pelekos, Lijian Jin, Maurizio S. Tonetti

**Affiliations:** 1Division of Periodontology and Implant Dentistry, Faculty of Dentistry, The University of Hong Kong, Hong Kong SAR, China; wangxz@hku.hk (X.W.); melissa.r.fok@gmail.com (M.R.F.); george74@hku.hk (G.P.); ljjin@hku.hk (L.J.); 2Shanghai PerioImplant Innovation Center, Department of Oral and Maxillofacial Implantology, Ninth People’s Hospital, Shanghai Jiao Tong University School of Medicine, Shanghai 200125, China; 3National Clinical Research Center of Stomatology, Shanghai 200125, China; 4European Research Group on Periodontology, 16149 Genova, Italy

**Keywords:** L-PRF, growth factors, release, first-order kinetics, in vitro, ex vivo

## Abstract

L-PRF is an autologous blood-derived biomaterial (ABDB) capable of releasing biologically active agents to promote healing. Little is known about its release profile of growth factors (GFs), cytokines, and MMPs. This study reported the in vitro and ex vivo release kinetics of GFs, cytokines, and MMPs from L-PRF at 6, 24, 72, and 168 h. The in vitro release rates of PDGF, TGF-β1, EGF, FGF-2, VEGF, and MMPs decreased over time with different rates, while those of IL-1β, IL-6, TNF-α, IL-8, and IL-10 were low at 6 h and then increased rapidly for up to 24 h and subsequently decreased. Of note, the release rates of the GFs followed first-order kinetics both in vitro and ex vivo. Higher rates of release were found ex vivo, suggesting that significant amounts of GFs were produced by the local cells within the wound. In addition, the half-life times of GFs locally produced in the wound, including PDGF-AA, PDGF-AB/BB, and VEGF, were significantly extended (*p* < 0.05). This work demonstrates that L-PRF can sustain the release of GFs and cytokines for up to 7 days, and it shows that the former can activate cells to produce additional mediators and amplify the communication network for optimizing the wound environment, thereby enhancing healing.

## 1. Introduction

Shortly after the discovery of the role of growth factors (GFs) in wound healing and their pioneering application in pre-clinical models, autologous blood-derived GF preparations have received great attention as an alternative to recombinant human preparations for promoting wound healing and tissue regeneration [1,2,3]. Systematic reviews on the effect of GFs show greater signals from autologous blood preparations than local applications of pharmaceutical preparations [1,4,5,6]. However, the evidence of their benefits is insufficient, and clinical equipoise remains for most clinical applications. 

Among the different preparations, leucocyte- and platelet-rich fibrin (L-PRF) is a second generation of platelet concentrates. Over the past decade, L-PRF has gained tremendous momentum and has been increasingly utilized for a variety of regenerative therapies, particularly in dentistry [7]. Systematic reviews show that the use of L-PRF could considerably improve good clinical outcomes in terms of soft and hard tissue healing [8,9,10]. 

L-PRF consists of platelets, leukocytes, and GFs harvested from blood, without the use of anticoagulants [7]. The lack of anticoagulant leads to the formation of thrombin, which successively activates most of the platelets and simultaneously triggers fibrin formation. L-PRF is obtained via a simplified preparation, without biochemical blood handling and follows only one centrifugation cycle [7]. During centrifugation, fibrinogen transforms into a dense fibrin network, which localizes in the middle portion of the tubes, between the red blood cells pellet (RBCs) and the upper acellular plasma. Most platelets (>90%) and more than half of the leukocytes (>60%) are concentrated in the intermediate layer located between the L-PRF clot and RBCs [11,12]. Platelets are activated during this process and various GFs, such as platelet-derived growth factor (PDGF), transforming growth factor-β1 (TGF-β1), epidermal growth factor (EGF), and vascular endothelial growth factor (VEGF), are released from the α-granules of platelets and embedded in the three-dimensional fibrin network. Indeed, L-PRF shows a prolonged release of these GFs from the fibrin matrix lasting for 7 days, due to its dense fibrin architecture that cannot completely dissolve in the culture medium. In addition to GF concentrates in L-PRF, inflammatory cytokines and chemokines (e.g., IL-1β, IL-4, IL-6, tumor necrosis factor–α (TNF-α), and IL-8) can also be released by both platelet α-granules and leukocytes [13].

Fibrin networks have been used as sustained/controlled release platforms/formulations for a variety of biologically active agents to promote wound healing and tissue regeneration [14]. These delivery systems aim to improve the performance of naturally occurring fibrin clots, but surprisingly, little is known about the release profile of autologous fibrin clot preparations. The release profile of GFs from L-PRF has been investigated in several studies using ELISA [12,15,16]. Knowledge about the release and the pharmacokinetic profile, however, remains incomplete. Additionally, there is little evidence of the relative proportion of GFs that are retained in the fibrin network as compared to those that are released and remain in aqueous solution. Furthermore, there is a lack of information in the literature regarding the parallel analysis of GFs and cytokines in the same sample. To address these limitations, this study investigated the in vitro and ex vivo release kinetics of GFs and cytokines from L-PRF at 6, 24, 72, and 168 h, and those observed in L-PRF construct exudate, using multiplex immunoassay. This notable approach allows for the simultaneous quantification of multiple markers providing unique information for a more complete understanding of L-PRF release profiles over 7 days. Therefore, the aim of this study was to describe the in vitro and ex vivo release profiles of a palette of 14 GFs and cytokines from L-PRF prepared from healthy subjects.

## 2. Materials and Methods

### 2.1. Blood Collection

Subjects were recruited from Prince Philip Dental Hospital between September 2019 and June 2021. These subjects participated in a clinical trial (clinicaltrials.gov NCT03985033) that was approved by the Institutional Review Board of the University of Hong Kong/Hospital Authority Hong Kong West Cluster (IRB Reference Number UW 19-306). All subjects provided written informed consent. The clinical data were reported separately [17]. In brief, a standard venipuncture was carried out on 18 subjects aged between 18 and 60 years old who had no history of recent aspirin intake or any medications that affect blood coagulation. Blood was collected from each subject in two types of tubes: silica-coated (to induce clotting) anticoagulant free blood collection tubes (Intraspin^®^; IntraLock™, Boca Raton, FL, USA) and blood collection tubes with EDTA. 

### 2.2. Preparation of L-PRF Clots

A total of 9 mL blood was collected in silica-coated tubes (IntraSpin^®^; IntraLock™, Boca Raton, FL, USA) for L-PRF clot preparation. L-PRF clots were prepared using the Choukroun’s protocol as described by Temmermann et al. [18]. The tubes were centrifuged after venipuncture at 2700 rpm (408 g) for 12 min at room temperature using a table centrifuge (Intraspin^®^; IntraLock™, Boca Raton, FL, USA). After centrifugation, a surgical tweezer was used to separate L-PRF clots from the tubes. L-PRF membranes for in vitro experiment were prepared by gravitational compression of L-PRF clots with weighted press against a perforated tray in a metal box (Xpression^TM^ Box; IntraLock™, Boca Raton, FL, USA) (Figure 1). Meanwhile, approximately 0.8 mL of the serum exudate expressed from each L-PRF clot was also collected for further analysis.

### 2.3. Plasma Preparation and Complete Blood Count

An amount of 6 mL of venous blood from each subject was collected in EDTA vacutainer and aliquoted for complete blood count and plasma preparation. Complete blood count (CBC) was performed using a Coulter counter in the Essence Medical Laboratory (Hong Kong). For plasma preparation, the whole blood was centrifuged for 10 min at 1500 g at 4 °C to remove cells. The supernatant was collected and stored at −80 °C until analysis.

### 2.4. In Vitro Release from L-PRF Membrane

To determine the release of biomarkers from L-PRF membrane at 6, 24, 72, and 168 h, L-PRF membranes were placed in a 24-well plate with 1 mL of sterile DMEM (Dulbecco’s Modified Eagle’s Medium) at 37 °C. At each time point, 1 mL of DMEM was collected and replaced with 1 mL of additional DMEM. The collected culture media were centrifuged at 10,000 rpm for 10 min at 4 °C to remove cellular elements and frozen thereafter at −80 °C until analysis.

### 2.5. Multiplex Assays

The concentrations of GFs, cytokines, and MMPs in plasma, exudate, and the collected culture media were evaluated using a multiplex immunoassay. A Milliplex^®^ MAP TGF-β1 Magnetic Bead Single Plex Kit (Merck & Co., Burlington, MA, USA) and Milliplex^®^ Customized 13-Plex Human Growth Factor/Cytokine/MMP Mag Kit (Merck & co., Burlington, MA, USA) was used to detect and quantify 14 biomarkers (TGF-β1, PDGF-AA, PDGF-AB/BB, EGF, FGF-2, VEGF, ANG-2, IL-1β, IL-6, IL-8, IL-10, TNF-α, MMP-1, and MMP-2) according to the manufacturer’s instructions. Briefly, the plate was washed with buffer. Standards or samples (25 μL) and assay buffer (25 μL) were added into each well. Antibody-immobilized-beads cocktail (25 μL) was resuspended by vortexing and added into each well of the microplate. The microplate was incubated with agitation on a plate shaker overnight at 4 °C. After washing away any unbound substances with wash buffer, 25 μL of detection antibody was added to each well and incubated for 1 h at room temperature. Then, 25 μL of streptavidin–phycoerythrin conjugate (Streptavidin-PE) was added and incubated for 30 min. Finally, after washing the plate, the beads were resuspended in 150 μL of sheath fluid and read within 90 min using the Bio-Plex^®^ 200 system (Bio-Rad Laboratories Inc., Hercules, CA, USA). Data were analyzed using a 5-parameter logistic (5PL) curve with Bio-Plex^®^ Manager 6.1 software (Bio-Rad Laboratories Inc., Hercules, CA, USA) and reported as ng/mL.

### 2.6. Ex Vivo Recovery

Raw data for ex vivo recovery were derived from the parent clinical trial registered in clinicaltrials.gov as NCT03985033. This trial reported growth factor and cytokine concentrations at 6, 24, 72, and 168 h in two similar tooth extraction wounds: one left to unassisted healing and the other receiving L-PRF [17]. In brief, wound fluid was sampled using sterile paper strips (Periopaper, Oraflow Inc., New York, NY, USA) from the wound edges and measured non-destructively using capacitance (Periotron 8010, Oraflow Inc., New York, NY, USA). As GFs and cytokines are also produced during the normal wound healing process, the effect of the added L-PRF was estimated by subtracting the values of the control sites at all time points.

### 2.7. Statistical Analysis

Statistical analyses were carried out using SPSS (Version 27.0, Chicago, IL, USA). This study was mainly descriptive. Normality distributions for all variables were checked with Shapiro–Wilk test. All continuous variables were expressed as mean ± SEM or median with interquartile ranges (Q1–Q3). Paired *t*-test or Wilcoxon matched-pairs signed-rank test was used to compare the difference between plasma and L-PRF exudate. A heatmap was generated using GraphPad Prism (Version 9.0, Dotmatics, Boston, MA, USA), and the average value was based on the mean of release rate. Pharmacokinetic parameters were estimated under the single-compartment open model assumption using the exponential one-phase-decay function of Prism GraphPad Version 9.0 (Dotmatics, Boston, MA, USA).

## 3. Results

### 3.1. Platelet and WBC Counts

A CBC analysis was performed on whole blood from each subject. As data were not normally distributed, median and interquartile ranges were reported for platelets and white blood cells (Table 1). All values were within the normal range.

### 3.2. Quantification of GFs, MMPs, and Cytokines in Plasma and L-PRF Exudate

Multiplex assay quantifications of GFs, cytokines, and MMPs from plasma and L-PRF exudate were determined for each patient (Table 2). Higher levels of TGF-β1, PDGF-AA, PDGF-AB/BB, EGF, VEGF, MMP-1 and MMP-2 were present in the PRF exudate when compared to those present in plasma from the same subject (Table 2). The levels of cytokines (IL-1β, IL-6, TNF-α, IL-10) and chemokines (IL-8) in plasma and L-PRF exudate were below the limit of detection of the assay in the majority of the 18 healthy donors, although significant higher levels of IL-8 and TNF-α were found in L-PRF exudate. 

### 3.3. Quantification of GFs, Cytokines, and MMPs In Vitro Released from L-PRF over Time

For each growth factor, cytokine, or MMP released from L-PRF, data were presented as rate of release as well as total release over time (Figure 2, Figure 3 and Figure 4). It was observed that the release rate of all the GFs decreased exponentially over time with slight variations among the GFs, and the results fit first-order kinetics (please see Section 3.6). 

The release rate of MMPs decreased over time. However, the release rate of cytokines (IL-1β, IL-6, TNF-α, IL-10) and a chemokine (IL-8) followed a different pattern that was largely consistent among the different molecules. Release rates were low at 6 h, increased rapidly between 6 and 24 h, and later decreased.

### 3.4. Heatmap Analysis

A heatmap was generated using the mean of total amounts of release per hour to show the changes in the release rate of GFs, cytokines, or MMPs over 168 h (Figure 5). Each analyte presented a specific release profile. For GFs and MMPs, the highest release was found at the first 6 h, followed by a reduction in concentrations, although with different rates of decrease, while the highest release rates for cytokines and the chemokine occurred from 6 to 24 h. 

### 3.5. Recovery of GFs in L-PRF Membrane and Exudate

Based on previous reports, the physiological levels of PDGF, EGF, FGF-2, and VEGF in platelets were 23 pg/10^6^ platelets, 2.5 pg/10^6^ platelets, 0.44 pg/10^6^ platelets, and 0.74 pg/10^6^ platelets, respectively [19,20]. The recovery of the four GFs from L-PRF membrane over 168 h and exudate were calculated. Higher amounts of PDGF, EGF, FGF-2, and VEGF were recovered from L-PRF membrane release compared to those detected in the exudate, confirming that most GFs remained trapped in the L-PRF (Table 3).

### 3.6. Comparison of In Vitro and Ex Vivo Patterns of Growth Factor Release

Some GFs, namely, TGF-β1, PDGF-AA, PDGF-AB/BB, EGF, FGF-2, and VEGF, followed a first-order release both in vitro and ex vivo (Figure 6, Table 4 and Table 5). A best-fit nonlinear regression was used based on the exponential one-phase-decay function. The half-life time of each growth factor was also calculated based on the equation in [21] to show the time required for the rate of release to decrease to half of its initial value (Table 4 and Table 5). Focusing on rate constants, half-life times of elimination, and estimated baseline rates of release, a few patterns emerged. PDGF-AA and PDGF-AB/BB showed similar baseline rates of release but had significantly smaller rate constants and longer half-life times ex vivo (*p* < 0.05). In contrast, VEGF showed a significantly higher half-life time for elimination ex vivo when compared to that in vitro (*p* < 0.05). TGF-β1, EGF, and FGF-2 displayed similar rate constants and half-life times, but they had higher rates of release ex vivo compared to in vitro, pointing to a predominant contribution from local production within the wound rather than release from the L-PRF formulation.

## 4. Discussion

This paper describes, for the first time, the pharmacokinetic patterns of the in vitro and ex vivo release of GFs, cytokines, and MMPs from L-PRF—a clinically applied autologous blood-derived construct for improved wound healing—through observing the local release of biologically active GFs using data from the same patient. Comparisons of the in vivo and ex vivo parameters revealed that the release of the GFs from L-PRF activates cells in the wound environment to produce additional mediators and amplify the communication network. 

Previous studies only reported the in vitro concentrations of specific GFs using single-analyte ELISA-based assays [12,15,16,22,23]. In this study, a multiplex immunoassay was applied to simultaneously quantify multiple biomarkers released from L-PRF in a single microplate well, and the method was based on a capture sandwich immunoassay. This method allows for the parallel analysis of GFs and cytokines in the same sample, thereby increasing the reliability of the results and reducing manipulation errors. It is interesting to note that the release of all GFs followed first-order kinetics both in vitro and ex vivo. However, higher rates of release were found ex vivo, indicating that significant amounts of GFs are produced by the local cells within a wound. The initial release of GFs from L-PRF may serve as a chemoattractant for neutrophils, macrophages, and fibroblasts and then further enhance their levels in a wound. In addition, the local production of GFs also extended the half-life times for PDGF-AA, PDGF-AB/BB, and VEGF. Such new findings demonstrate that L-PRF could not only release GFs per se but also promote local host cells to produce more GFs in vivo, thereby providing indirect evidence of the bioactivity of the released mediators.

GFs and cytokines play an important role in wound healing regulation due to their capacity to stimulate cell migration, proliferation, and differentiation as well as their involvement in complex cellular signaling networks for coordinating multiple cells [24]. Several biomolecules have already been investigated in L-PRF by other groups using elution, such as TGF-β1 [12,15,16,22,23], PDGF-AA [15,25], PDGF-AB/BB [12,15,16,22,23], EGF [15], VEGF [12,15,16,22,23], and MMP-1 [12], which reflects the change in the concentration of GFs in the culture medium. A recent study showed that the total amounts of GFs released during the first 7 days were always higher than the total quantities after forcible extraction from L-PRF clots after preparation [23]. This finding suggests that cells trapped within L-PRF (mainly leukocytes) can contribute to the production of GFs. Physiological levels of PDGF, VEGF, and FGF-2 in platelets are 23 ± 6 pg/106 platelets, 0.74 ± 0.37 pg/106 platelets, and 0.44 ± 0.15 pg/106 platelets, respectively [19]. Such values clearly indicate that large amounts of GFs can be released from platelets. Of the total reservoir of GFs present in platelets, a portion may be lost in the supernatant during L-PRF preparation due to early platelet degranulation and activation, while a part may be loosely associated with the three-dimensional clot matrix and may be removed during squeezing of the L-PRF clot in the exudate. Another fraction may be more tightly embedded into the fibrin mesh but still be present within platelets, and it could be slowly released over time. In this study, the total recovery of GFs ranged between 13% (VEGF) and 87% (PDGF). While these calculations used historical data and thus are inherently imprecise, the fact that a large fraction of the total GFs was released slowly from L-PRF speaks for the effectiveness of the preparation method and its biologic potential during local applications for the promotion of wound healing.

Notably, within 7 days, the accumulated concentrations of released GFs were higher than those in plasma, except for ANG-2. Furthermore, the three-dimensional polymer structure of the fibrin matrix prevented the degradation of platelets so that the GFs could be released gradually. Castro et al. reported the TGF-β1, PDGF-AB, and VEGF release profiles by using ELISA detection methods at five time intervals (0 to 4 h, 4 to 24 h, 24 to 72 h, 72 to 168 h, and 168 to 336 h), showing a continuous release of these GFs for over 336 h [23]. Their results are consistent with those of the present study. However, it is difficult to compare absolute concentrations due to methodological differences, such as the volume of culture medium and the detection approach (ELISA vs. multiplex immunoassay).

In this study, it was interesting to observe that Choukroun’s L-PRF could also gradually release cytokines and chemokines over time. High concentrations of IL-1β, TNF-α, IL-6, IL-10, and IL-8 were released from L-PRF after 6 h, and they were likely released step-by-step during the experiment by the leukocytes embedded in the L-PRF fibrin matrix. The presence of leukocytes in platelet concentrates and the consequent release of inflammatory mediators may be beneficial for wound healing. Previous studies have shown that the high release of inflammatory cytokines, including IL-1β and TNF-α, can induce the proliferation and differentiation of osteoblasts and osteoclasts [26]. Also, it was found that IL-1β released by L-PRF plays an active role in tissue repair, which may promote the migration of mesenchymal stem cells and human endothelial cells [27]. Moreover, high amounts of cytokines can stimulate the defense mechanism, which is important in cases of wound infection.

For clinical applications, L-PRF clots prepared using the standard procedure are usually squeezed to obtain L-PRF membranes for clinical practice, with the PRF exudate being discarded. It was reported by Castro et al. (2019) that only 6% of platelets and 0.9% of leukocytes were present in L-PRF exudate, while most of them were trapped in the fibrin matrix [23]. Our results showed that the levels of GFs recovered in exudate were much lower than the accumulated amounts released from the L-PRF membrane, indicating very little degranulation from platelets during the clotting/preparation of the L-PRF. However, it was interesting to find that higher levels of PDGF-AA, PDGF-AB/BB, TGF-β1, EGF, and VEGF were present in the L-PRF exudate than those present in plasma, which might suggest that a small part of molecular content was released from L-PRF after squeezing. Given the presence of GFs in L-PRF exudate, it could be mixed with bone grafts, forming a bioactive scaffold to promote wound healing. Witek et al. reported that L-PRF exudate promoted bone regeneration when incorporated with the PLGA scaffold as a grafting material [28].

To reduce inter-individual variability, all the subjects recruited in this study were healthy. The number of platelets and WBCs were within the physiologic ranges. GFs, chemokines, cytokines, and MMPs in plasma were also reported in our study. All the cytokines and the chemokine were present in very low amounts, which is consistent with findings reported by Biancotto et al., indicating the same concentration scale [29].

Although all the subjects were healthy, the individual variability between subjects was still large, as the characteristics of L-PRF can be affected by patient heterogeneity in terms of age, gender, and healing capabilities [30]. Yajamanya et al. found that the number of platelets and white blood cells decreased with age, which might affect the levels of GFs and cytokines [11]. Due to these limitations, further studies should increase sample sizes to minimize variability. Moreover, as L-PRF could continuously release growth factors for up to 14 days, a long observation period is necessary to reveal the comprehensive releasing profiles of growth factors from L-PRF.

## 5. Conclusions

This study indicated that the release of all GFs that were investigated followed first-order kinetics both in vitro and ex vivo. Only a small fraction of platelet-associated GFs, such as PDGF, was released during the clotting and preparation of the L-PRF. The current findings on the continuous release of GFs and cytokines from L-PRF membrane for up to 7 days and indirect evidence of in vivo amplification of cellular communication networks highlight that L-PRF can be employed in medical and dental fields for enhancing tissue regeneration.

## Figures and Tables

**Figure 1 cells-11-02089-f001:**
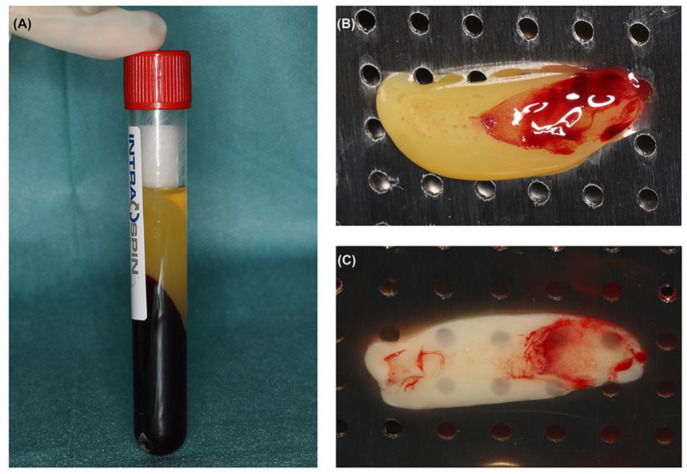
L-PRF preparation. (**A**) Tubes after centrifugation process. Content in three layers: red blood cells, L-PRF clot, and acellular plasma—from the lower end to the top of the tube; (**B**) L-PRF clot; (**C**) L-PRF membrane.

**Figure 2 cells-11-02089-f002:**
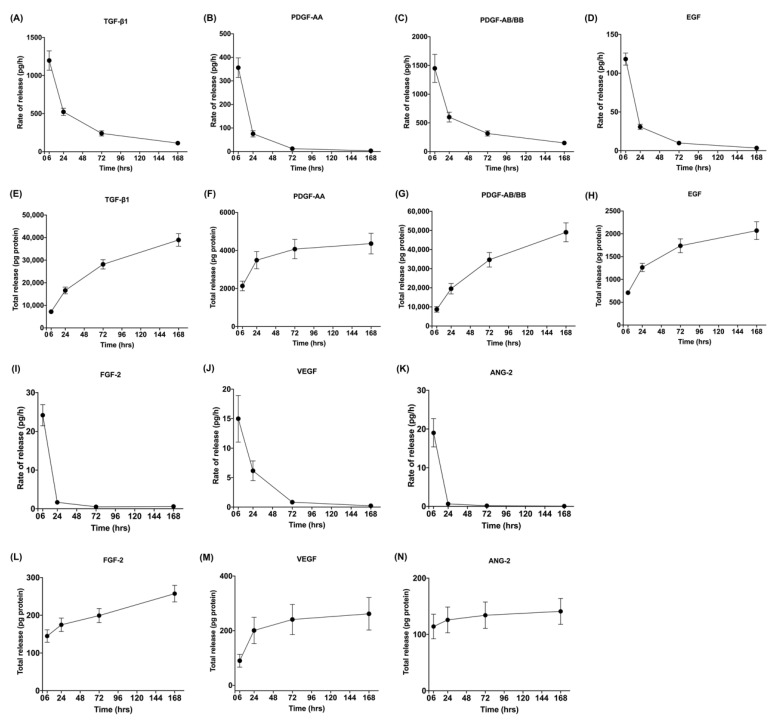
Rate of release and total release of GFs. The figure shows the rate of release and total release of the tested GFs from the L-PRF membrane up to 168 h. Panels (**A**,**E**) display TGF-β1, (**B**,**F**) PDGF-AA, (**C**,**G**) PDGF-AB/BB, (**D**,**H**) FGF-2, (**I**,**L**) EGF, (**J**,**M**) VEGF, and (**K**,**N**) ANG-2 for rate of release and total release, respectively. Mean ± SEM is displayed at each time point.

**Figure 3 cells-11-02089-f003:**
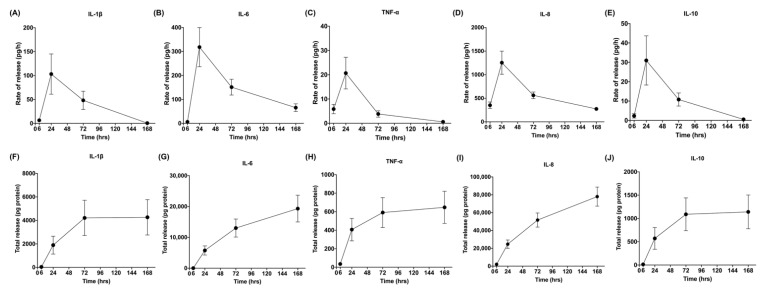
Rate of release and total release of cytokines. The figure shows the rate of release and total release of the tested cytokines from the L-PRF membrane up to 168 h. Panels (**A**,**F**) display IL-1β, (**B**,**G**) IL-6, (**C**,**H**) TNF-α, (**D**,**I**) IL-8, (**E**,**J**) and IL-10 for rate of release and total release, respectively. Mean ± SEM are displayed at each time point.

**Figure 4 cells-11-02089-f004:**
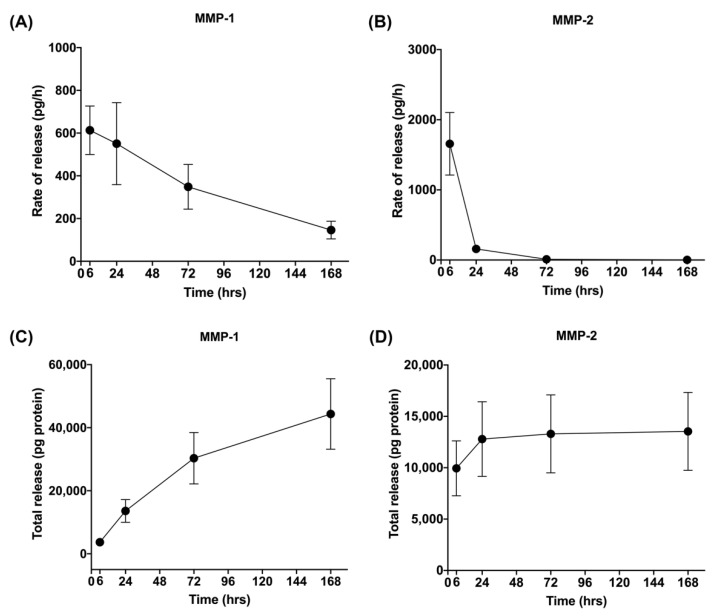
Rate of release and total release of matrix metalloproteinases (MMPs). The figure shows the rate of release and total release of the tested matrix metalloproteinases (MMPs) from the L-PRF membrane up to 168 h. Panels (**A**,**C**) display MMP-1 and (**B**,**D**) MMP-2 for rate of release and total release, respectively. Mean ± SEM is displayed at each time point.

**Figure 5 cells-11-02089-f005:**
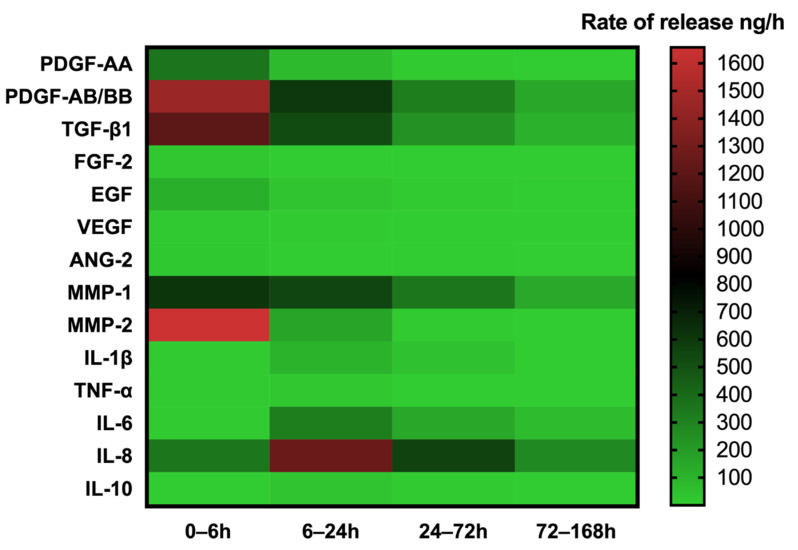
Heatmap of the rate of release of GFs, cytokines, and MMPs up to 168 h. The heatmap represents the changes in the mean of release rate of GFs, cytokines, or MMPs over 168 h. The release rate is indicated according to the color scale representing different levels of rate of release per hour (green—lowest, red—highest rate of release).

**Figure 6 cells-11-02089-f006:**
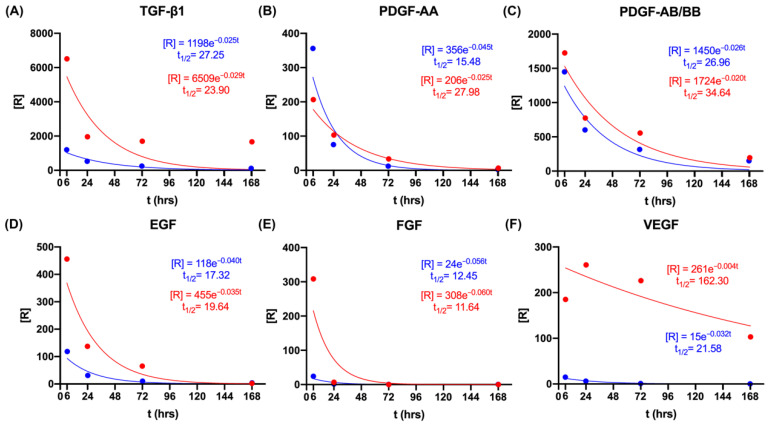
One-phase exponential decay of GFs in vitro and ex vivo. The figure shows one-phase exponential decay of (**A**) TGF-β1, (**B**) PDGF-AA, (**C**) PDGF-AB/BB, (**D**) EGF, (**E**) FGF-2, and (**F**) VEGF in vitro (blue) and ex vivo (red). Significant differences in half-life times were observed for PDGF-AA (*p* = 0.004), PDGF-AB/BB (*p* = 0.028), and VEGF (*p* = 0.033) between the in vitro and ex vivo cases. [R]: rate of release; t_1/2_: half-life time.

**Table 1 cells-11-02089-t001:** Platelet and WBC counts in whole blood.

	Concentration (10^9^/L) Median (Q1–Q3)	Adult Reference Ranges (10^9^/L)
Platelet	295.5 (240.75–324.75)	150–400
White blood cell	6.00 (5.13–7.68)	4.00–11.00

**Table 2 cells-11-02089-t002:** GFs, cytokines, and MMPs concentrations in plasma and L-PRF exudate.

	Blood Plasma Median (Q1–Q3)	L-PRF Exudate Median (Q1–Q3)	Fold Changes (L-PRF Exudate/Plasma)	*p* Value
TGF-β1 (pg/mL)	4574.25 (2393.63–7234.80)	14,169.00 (10,737.30–17,356.95)	3.10	<0.0001
PDGF-AA (pg/mL)	309.30 (128.48–417.13)	1507.81 (1087.01–2063.21)	4.87	<0.0001
PDGF-AB/BB (pg/mL)	2701.45 (1220.95–4043.44)	6676.02 (6245.01–8360.68)	2.47	<0.0001
EGF (pg/mL)	5.75 (2.90–7.77)	20.24 (5.43–28.51)	3.52	0.0007 ^#^
FGF-2 (pg/mL)	66.72 (48.36–79.96)	42.34 (17.83–68.84)	0.63	0.0665 ^#^
VEGF (pg/mL)	4.90 (4.90 *–21.38)	41.43 (4.90 *–89.75)	8.46	0.0046 ^#^
ANG-2 (pg/mL)	618.15 (444.69–1025.74)	741.31 (525.68–1208.38)	1.20	0.1084 ^#^
MMP-1 (pg/mL)	137.02 (77.22–199.98)	633.66 (416.20–980.59)	4.62	<0.0001 ^#^
MMP-2 (pg/mL)	9830.76 (8283.60–12,369.32)	10,864.02 (8925.92–15,183.59)	1.11	0.0004
IL-1β (pg/mL)	0.66 (0.10 *–1.32)	0.10 (0.10 *–1.47)	0.15	0.1733 ^#^
IL-6 (pg/mL)	1.46 (0.21 *–6.27)	2.39 (0.21 *–8.33)	1.64	0.6698 ^#^
TNF-α (pg/mL)	8.04 (6.56–10.12)	9.99 (6.44–14.63)	1.24	0.0026 ^#^
IL-8 (pg/mL)	5.04 (2.91 *–10.09)	8.79 (5.00–13.36)	1.74	0.0019 ^#^
IL-10 (pg/mL)	3.26 (2.54–4.36)	2.31 (1.62 *–3.26)	0.71	0.003

Paired *t*-test (for normally distributed biomarkers). ^#^ Wilcoxon matched-pairs signed-rank test (for not normally distributed parameters). * Value at the limit of detection of the assay.

**Table 3 cells-11-02089-t003:** GF recovery from L-PRF and exudate.

	PDGF (PDGF-AA, PDGF-AB/BB)	EGF	FGF-2	VEGF
Physiological amount of GF per 10^6^ platelets (pg)	23	2.5	0.44	0.74
Estimated amount of GF per 9 mL of blood (pg, Q1–Q3)	61,168.5 (49,835.3–67,223.3)	6648.75 (5416.9–7306.9)	1170.18 (953.4–1286.0)	1968.03 (1603.4–2162.8)
Total amount released from L-PRF (pg)	53,412.35	2070.12	257.34	261.67
% Recovery from L-PRF membrane	87.32%	31.14%	21.99%	13.30%
% Recovery from exudate	10.70%	0.24%	2.89%	1.68%

**Table 4 cells-11-02089-t004:** Best-fit values, 95% CIs, and goodness of fits of the exponential decay of each growth factor in vitro.

	TGF-β1	PDGF-AA	PDGF-AB/BB	EGF	FGF-2	VEGF
Best-fit values						
Y0	1198	356	1450	118	24	15
K	0.025	0.045	0.026	0.040	0.056	0.032
Half Life	27.25	15.48	26.96	17.32	12.45	21.58
Tau	39.31	22.33	38.89	24.99	17.96	31.13
95% CI (profile likelihood)						
K	0.011 to 0.059	0.016 to 0.128	0.010 to 0.067	0.015 to 0.109	0.015 to 0.211	0.016 to 0.067
Half Life	11.66 to 61.40	5.40 to 43.81	10.35 to 67.20	6.39 to 46.76	3.29 to 47.30	10.42 to 43.68
Tau	16.82 to 88.58	7.79 to 63.21	14.93 to 96.94	9.21 to 67.45	4.74 to 68.24	15.03 to 63.02
Goodness of Fit						
R squared	0.92	0.89	0.90	0.90	0.83	0.94

Y0: the initial amount. Tau: the time constant. K: the rate constant.

**Table 5 cells-11-02089-t005:** Best-fit values, 95% CI and goodness of fit of the exponential decay of each growth factor ex vivo.

	TGF-β1	PDGF-AA	PDGF-AB/BB	EGF	FGF-2	VEGF
Best-fit values						
Y0	6509	206	1724	455	308	261
K	0.029	0.025	0.020	0.035	0.060	0.004
Half Life	23.90	27.98	34.64	19.64	11.64	162.3
Tau	34.49	40.37	49.97	28.33	16.79	234.2
95% CI (profile likelihood)						
K	0.004 to 0.174	0.014 to 0.045	0.008 to 0.060	0.013 to 0.095	0.015 to 0.252	0.001 to 0.014
Half Life	3.99 to 160.90	15.47 to 49.34	11.64 to 91.04	7.32 to 51.38	2.75 to 47.39	50.03 to 31961
Tau	5.76 to 232.20	22.32 to 71.18	16.80 to 131.30	10.57 to 74.12	3.96 to 68.36	72.18 to 46111
Goodness of Fit						
R squared	0.63	0.96	0.87	0.90	0.82	0.48

Y0: the initial amount. Tau: the time constant. K: the rate constant.

## Data Availability

Original data are stored in the system of the graduate school of the University of Hong Kong. Original data will be available upon reasonable request.

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
