# Peer review of "In Vitro and Ex Vivo Kinetic Release Profile of Growth Factors and Cytokines from Leucocyte- and Platelet-Rich Fibrin (L-PRF) Preparations"

_cells, 2022, doi:10.3390/cells11132089_

Round 1

Reviewer 1 Report

In the submitted paper “In vitro and ex vivo Kinetic Realese Profile of Growth Factors and Cytokines from Leucocyte- and Platelet-Rich Fibrin (L-PRF) Preparations” the authors investigated the in vitro and ex vivo release kinetics of 16 growth factors, cytokines and MMPs from L-PRF at 6, 24, 72 and 168 hours.

The authors should improve the introduction section by mentioning the clinical applications of L-PRF in regenerative therapies, particularly in dentistry. They should also emphasized that L-PRF accelerate the wound healing process and achieve good results in terms of bone quality and soft tissue healing. Please consider the papers PMID: 29870308, 30072300, 31124821, 32577315. Then renumber citation in the text and in the references section.

Furthermore, in several part of this paper the authors stated that L-PRF can sustain release growth factors and cytokines up to 7 days (line 24-25, line 51, line 299). However, in literature a slow release of growth factors from PRF has been reported for up to 14 days. In the discussion section the authors should analyze this concept. Please consider the paper PMID: 32290550. Then renumber citation in the text and in the references section.

The paper is of interest for scientific literature but it needs references improvement and minor revisions before publication.

Reviewer 2 Report

In the submitted paper “In vitro and ex vivo Kinetic Realese Profile of Growth Factors and Cytokines from Leucocyte- and Platelet-Rich Fibrin (L-PRF) Preparations” the authors described for the first  the in vitro and ex vivo kinetic release profile of  different growth factors and cytokines from human L-PRF and the results showns that L-PRF is able to stimulate further release of these factors up to 7 days by local cells in wounds and thereby enhancing healing.

The work is well structured and well written and I have very few minor comments about it:

- introduction: add some references of human clinical application of L-PRF.

- M and M: the information on the ex vivo trial is not exhaustive and must be reported briefly, not only by citing the article of origin. Heatmap generation software should be described because it is not clear to all readers.

- Figures 1,2 and 3 are really very small and difficult to read compared to the text. it would be better to enlarge it or make it more readable in some other way.

Reviewer 3 Report

This is a valuable and unique study as it has, for the first time, compared the in vitro and ex vivo release of growth factors and cytokines from an autologous biomaterial PLRF. For the latter, the data from a clinical trial were used. Significantly, the results have shown that in vivo, PLRF is able to stimulate further release of these molecules by local cells in the wound,  therefore its beneficial effect is not simply related to a burst release from the material itself but is amplified locally.  Different release kinetics for MMPs, cytokines and GFs are also interesting and shed light on the role of leukocytes and biomaterial matrix in the preparations. I suggest several comments to further  improve the study:

Major comments:

1. The authors should add an image or a diagramm on how LPRF membranes were made (for non-dental audience). What was their size? Was only one membrane made from 9 ml sample? What was the volume of the exudate?

2. More information in M&M section is required for the sample processing from the clinical trail. Were the measurements done in blood plasma or wound plasma? The latter seems to be correct, but it has not been made explicit in the M&M section.

3. Table 2. The authors should add 'Fold differences' column, and sort the data based on fold differences. That way, the reader can easier determine which molecules showed the largest differences.

4. Heatmap generation software should be briefly described in the M&M section. Were the average values (indicated by colour) based on the means or medians?

5. Table 3. Were the estimated amounts of GF based on the median number of platelets from the study group, as presented in Table 1? For a fuller picture, the authors should include the estimates for the lower and upper quartiles (since they themselves noted high heterogeneity within the cohort).

Minor comments.

1. Sentence: The levels of cytokines (IL-1β, IL-6, TNF-α, IL-10) and chemokines (IL-8) in plasma and L-PRF exudate were below the limit of detection of the assay in the majority of the 18 healthy-donors, although significant higher levels of IL-8 and TNF-α were found in L-PRF exudate (in some samples??)

Round 2

Reviewer 1 Report

Good job and congratulation for your valuabel work!

Reviewer 3 Report

   Happy with the revision